# Xylitol Inhibits Growth and Blocks Virulence in *Serratia marcescens*

**DOI:** 10.3390/microorganisms9051083

**Published:** 2021-05-18

**Authors:** Ahdab N. Khayyat, Wael A. H. Hegazy, Moataz A. Shaldam, Rasha Mosbah, Ahmad J. Almalki, Tarek S. Ibrahim, Maan T. Khayat, El-Sayed Khafagy, Wafaa E. Soliman, Hisham A. Abbas

**Affiliations:** 1Department of Pharmaceutical Chemistry, Faculty of Pharmacy, King Abdulaziz University, Jeddah 21589, Saudi Arabia; ankhayyat@kau.edu.sa (A.N.K.); ajalmalki@kau.edu.sa (A.J.A.); tmabrahem@kau.edu.sa (T.S.I.); mkhayat@kau.edu.sa (M.T.K.); 2Department of Microbiology and Immunology, Faculty of Pharmacy, Zagazig University, Zagazig 44519, Egypt; hishamabbas2008@gmail.com; 3Department of Pharmaceutical Chemistry, Faculty of Pharmacy, Kafrelsheikh University, Kafr El-Sheikh 33516, Egypt; dr_moutaz_986@yahoo.com; 4Infection Control Unit, Zagazig University Hospitals, Zagazig University, Zagazig 44519, Egypt; rashamosbah6@gmail.com; 5Faculty of Oral and Dental Medicine, Ahram Canadian University, Giza Governorate 12573, Egypt; 6Department of Pharmaceutical Organic Chemistry, Faculty of Pharmacy, Zagazig University, Zagazig 44519, Egypt; 7Department of Pharmaceutics, College of Pharmacy, Prince Sattam Bin Abdulaziz University, Al-Kharj 11942, Saudi Arabia; e.khafagy@psau.edu.sa; 8Department of Pharmaceutics and Industrial Pharmacy, Faculty of Pharmacy, Suez Canal University, Ismailia 41552, Egypt; 9Department of Biomedical Science, Faculty of Clinical Pharmacy, King Faisal University, Alhofuf, Al-Ahsa 36362, Saudi Arabia; wafaaezz2006@yahoo.com; 10Department of Microbiology and Biotechnology, Faculty of Pharmacy, Delta University for Science and Technology, Mansoura 11152, Egypt

**Keywords:** *Serratia marcescens*, quorum sensing, virulence, xylitol

## Abstract

*Serratia marcescens* is an opportunistic nosocomial pathogen and causes wound and burn infections. It shows high resistance to antibiotics and its pathogenicity is mediated by an arsenal of virulence factors. Another therapeutic option to such infections is targeting quorum sensing (QS), which controls the expression of different *S. marcescens* virulence factors. Prevention of QS can deprive *S. marcescens* from its bacterial virulence without applying stress on the bacterial growth and facilitates the eradication of the bacteria by immunity. The objective of the current study is to explore the antimicrobial and antivirulence activities of xylitol against *S. marcescens*. Xylitol could inhibit the growth of *S. marcescens*. Sub-inhibitory concentrations of xylitol could inhibit biofilm formation, reduce prodigiosin production, and completely block protease activity. Moreover, xylitol decreased swimming motility, swarming motility and increased the sensitivity to hydrogen peroxide. The expression of *rsmA*, *pigP*, *flhC*, *flhD fimA*, fimC, *shlA bsmB*, and *rssB* genes that regulate virulence factor production was significantly downregulated by xylitol. In silico study showed that xylitol could bind with the SmaR receptor by hydrophobic interaction and hydrogen bonding, and interfere with the binding of the natural ligand with SmaR receptor. An in vivo mice survival test confirmed the ability of xylitol to protect mice against the virulence of *S. marcescens*. In conclusion, xylitol is a growth and virulence inhibitor in *S. marcescens* and can be employed for the treatment of *S. marcescens* wound and burn infections.

## 1. Introduction

*Serratia marcescens* is an environmental rod-shaped gram-negative bacterium that belongs to the Enterobacteriaceae family. *S. marcescens* is an opportunistic microbe that can cause several nosocomial infections [1]. Moreover, *S. marcescens* can cause wound and soft tissue infections [2]. Considerably, *S. marcescens* is responsible for 11% of open burn-related surgical wound infections [3], and is one of the gram-negative bacteria that causes invasive burn wound infections [4]. *S. marcescens’* pathogenicity is attributed to its arsenal of virulence factors including protease, nuclease, lipase and hemolysin in addition to its ability to swarm and swim and to form biofilm [5]. *S. marcescens* produces a red pigment, prodigiosin, which possesses different activities against fungi, bacteria and protozoa [6,7]. Moreover, prodigiosin has immunosuppressive effects [8].

The treatment of *S. marcescens* infections is hindered by its multidrug-resistant nature. It has showed resistance to fluoroquinolones, aminoglycosides and β-lactams [9]. Furthermore, *S. marcescens* can adhere to medical instruments forming biofilms. Biofilm formation adds to the problem of antibiotic resistance by forming a protective shield for microorganisms against antibiotics [1]. Using antibiotics poses a selective growth stress on bacteria that leads to the emergence of bacterial resistance. This is why there is an increasing demand to develop alternative strategies. One of them is targeting the intercellular communication system, quorum sensing (QS), which helps bacteria to survive in stressed conditions [5]. QS depends on secretion of signaling molecules or auto-inducers that regulate the expression of virulence genes in bacterial densities when they reach a certain threshold of concentration [7,10,11]. In *S. marcescens*, QS regulates the expression of several virulence factors including nuclease, hemolysin, lipase, protease, prodigiosin, biofilm formation and motility [12]. QS inhibitors pave the way to a new trend in treating *S. marcescens* infections, either alone or in combination with antibiotics [5,13]. This approach does not affect bacterial growth, and development of resistance is much less likely to occur. Moreover, inhibition of QS can enhance the immune response against bacteria [14,15].

The use of sugars such as sucrose and hypertonic glucose has been reported as a bacterial growth inhibitor. The application of sugars to the wound provides a local osmotic effect that enhances the formation of granulation tissue, reduces edema in wounds, and leads to improvement of microcirculation of the wound. As a result, the healing time of the patient is reduced and the needed medical care is lessened [16,17,18]. Xylitol is a sugar alcohol that was previously found to have antimicrobial activity against *Streptococcus pneumoniae* and *Streptococcus mutants* [19]. In addition, xylitol could inhibit the adhesion of many bacteria [20]. The application of xylitol in wound care was reported in an in vitro Lubbock Chronic Wound Biofilm model, where the growth of Staphylococcus aureus, *Pseudomonas aeruginosa*, and *Enterococcus faecalis* was reduced by the use of 2%, 10%, and 20% xylitol in water and biofilm formation was completely blocked by 20% xylitol [21]. Moreover, combining 2% lactoferrin with 5% xylitol decreased the biofilm formation in methicillin-resistant *S. aureus* and *P. aeruginosa* as compared to base wound dressings alone [22]. Furthermore, the negative dissolution energy of xylitol exerts a cooling effect on the inflamed tissues [22,23].

In our previous study, xylitol could inhibit biofilms of *P. aeruginosa* clinical isolates [24]. Interestingly, xylitol was found to possess antivirulence activity against the model bacterium *Chromobacterium violaceum* CV026 by affecting QS [23]. However, the activity of xylitol on the growth, QS and virulence in *S. marcescens* has not been previously studied, to our knowledge. In this study, the antimicrobial and antivirulence effects of xylitol on *S. marcescens* were investigated.

## 2. Materials and Methods

### 2.1. Chemicals and Microbiological Media

All chemicals were of pharmaceutical grade; these include microbiological media, Tryptone soya broth and agar, Luria-Bertani (LB) broth and agar, and Mueller Hinton broth. Xylitol was ordered from Sigma-Aldrich (St. Louis, MO, USA).

### 2.2. Bacterial Strains

The *S. marcescens* clinical isolate used in the current study was obtained from a patient that was admitted to the Intensive Care Unit at Zagazig University Hospital from a surgical wound infection. This isolate was fully identified by the MALDI-TOF apparatus [7,10,25].

### 2.3. Determination the Minimum Inhibitory Concentration (MIC) of Xylitol

To determine the MIC of xylitol, the broth microdilution method was employed (CLSI, 2015), and MIC was determined as the least concentration of xylitol with which no visible bacterial growth was observed.

### 2.4. Evaluation of Biofilm Production

For assessment of biofilm production, *S. marcescens* overnight cultures were cultivated, diluted with tryptone soya broth (TSB) and the optical density was adjusted to a cell density of 1 × 106 CFU/mL [10,26]. Aliquots of 200 μL of the prepared bacterial suspension were transferred in sterile 96-well polystyrene microplates and incubated overnight at 37 °C. The non-adherent planktonic cells were aspirated and the wells were washed. The adherent cells were fixed with methanol (99%) for 25 min, and stained with crystal violet (1%) for 25 min. The unattached dye was washed off, and the plates were air-dried. The bounded crystal violet dye was extracted by 33% glacial acetic and absorbance was measured with 95% ethanol. The test was repeated in triplicate, the mean optical densities were calculated at a wavelength of 590 nm, and the cut-off optical density (ODc) was calculated. The cut-off OD (ODc) is defined as three times standard deviations above the mean OD of the negative control. The tested isolate was categorized into one of four groups; non-biofilm forming (OD ≤ ODc), weak-biofilm forming (OD > ODc, but ≤ 2× ODc), moderate-biofilm forming (OD > 2× ODc, but ≤ 4× ODc), or strong-biofilm forming (OD > 4× ODc).

### 2.5. Biofilm Inhibition Assay

The xylitol inhibitory activity on the biofilm formation was assessed as indicated formerly [15,25,27,28]. The xylitol inhibitory effects were evaluated at different concentrations (5% and 10%) and the biofilm inhibition was calculated and presented as a percentage of the control free from xylitol.

Moreover, the biofilm inhibition by xylitol was visualized under light microscope [7,29]. The formed biofilms on cover slips in presence or absence of xylitol (5% or 10%) were stained with crystal violet and imaged under light microscope.

### 2.6. Prodigiosin Inhibition Assay

The ability of xylitol to interfere with the prodigiosin production of *S. marcescens* was evaluated as previously shown [7,10,25]. The prodigiosin dye was extracted and acidified from bacterial cultures that were treated with or without 5% or 10% xylitol, and the absorbance was estimated at 534 nm via Biotek Spectrofluorometer (Biotek, Winooski, VT, USA). The prodigiosin inhibition was expressed as a percentage of the untreated xylitol-free control.

### 2.7. Protease Assay

The proteolytic activity of the tested *S. marcescens* was assessed in the absence or presence of 5% or 10% xylitol by the skim milk agar method [7,15]. The supernatants containing extracellular protease collected from treated bacterial cultures provided with or without xylitol were collected and added to wells in 5% skim milk LB agar plates. After incubation, the formed clear zones due to protease activity were measured. The percentage of protease inhibition was shown as a percentage of the xylitol-free control.

### 2.8. Assay of Bacterial Motilities

*S. marcescens* motility assays were performed in the presence or absence of 5% or 10% xylitol [7,15,25,30]. The LB 0.3% or 0.5% agar plates with or without 5% or 10% xylitol were prepared, and centrally cultured with *S. marcescens* for swimming or swarming motilities assays, respectively. The swimming or swarming zones in xylitol containing plates were measured in mm and presented in comparison to control xylitol-free plates.

### 2.9. Oxidative Stress Assay

In order to investigate the ability of xylitol to restrict the *S. marcescens* resistance to oxidative stress, the oxidative stress was assessed in presence of xylitol [15,31]. Tryptone soya agar (TSA) plates with xylitol (5% and 10%) and control plates without xylitol were prepared. *S. marcescens* isolate was grown overnight in TSB and 100 μL from the culture were spread on the surface of the plates. Ten microliters of 3% H_2_O_2_ were added to sterile filter-paper disks on the surface of plates. After overnight incubation, the inhibition zones due to H_2_O_2_ were measured.

### 2.10. Quantification of Virulence Genes’ Expressions

In order to explore the influence of xylitol on expression of virulence genes, the RNA was extracted from *S. marcescens* cultures treated or untreated with xylitol (5%) using RNeasy Mini kit (Qiagen, Hilden, Germany). The extracted RNA concentrations were quantified by NanoDropTM 2000 (Thermo Fisher Scientific, Waltham, MA, USA). A Reverse Transcriptase kit (Thermo Fisher Scientific, Waltham, MA, USA) was employed to synthesize cDNA from extracted RNA samples. The cDNA amplification was performed with the SYBR Green Quantitative RT-qPCR Kit (Thermo Fisher Scientific, Waltham, MA, USA). The expression levels of virulence genes *fimA*, *fimC*, *rssB*, *bsmB*, *rsmA*, *shlA*, *pigB*, *flhC* and *flhD* were standardized in relation to the critical threshold (CT) mean values of *rplU* as a housekeeping gene by the 2^−∆∆Ct^ method [32,33,34]. The designed genes’ primers and PCR cycles used in this study have been designated formerly [7,25].

### 2.11. Mice Survival Test

The in vivo survival model was used to assess xylitol’s protective activity against *S. marcescens* virulence [7,15,33,35]. Approximately, 1 × 10^8^ CFU/mL *S. marcescens* treated with or without sub-MIC of xylitol (100 mg/mL) were suspended in phosphate-buffered saline. Twenty healthy female albino mice (*Mus musculus*) with the same age, 3-weeks old, and same weight were distributed in 4 groups. In the first group, mice were intraperitoneally injected with 100 μL of xylitol-treated *S. marcescens*. In the second positive control group, mice were injected with untreated *S. marcescens*. In the third and fourth negative control groups, mice were either injected with 100 μL of sterile PBS or kept uninoculated. All mice were kept in safe and suitable cages with normal feeding and aeration in the Faculty of Pharmacy animal house, Zagazig University, Egypt. The mice survival in all groups was noted every day for 5 consecutive days.

### 2.12. In-Silico Analysis of Binding to SmaR Receptor

The ability of xylitol to bind with the protein receptor SmaR was investigated by docking analysis. The SmaR protein 3D structure was created by SWISS-MODEL Server [36] using the crystal structure of the bacterial quorum-sensing transcription factor as a template (PDB code 1L3L) [37]. The study was carried out on xylitol and the co-crystalized natural ligand, C4HSL, in the receptor’s active site using AutoDock Vina [38]. Ligand structures were sketched into Marvin Sketch V18.23.0 (Marvin Sketch, 2019), and the most energetically favored conformer was presented in format (*.pdb). All the rotatable bonds in ligands that were set to be flexible and the package of AutoDockTools was employed to detect Gasteiger atomic partial charges [39]. For preparation of the receptor, hydrogen atoms were included, Gasteiger atomic partial charges were determined, and the receptor and ligands were changed to the PDBQT format. In the configuration of AutoDock Vina, the parameter num modes were adjusted to 10 and exhaustiveness to 10. The grid box center (x = 20.67, y = 20.59 and z = 20.06) with size (x = 13, y = 13, z = 13) was used to recognize the docking active site. Pymol (PyMOL Molecular Visualization System 2019) was used for 3D visualization and the 2D graphic presentation was created using LigPlot+ V1.4.5 [40].

### 2.13. Statistical Analysis

All assays were conducted in triplicate and the one-way ANOVA test (Graph Pad Prism 8) was employed for evaluation of the significance of xylitol’s inhibitory activities on *S. marcescens* virulence factors. Statistical significance was considered when *p* values < 0.05.

## 3. Results

### 3.1. Determination of MIC

Xylitol could inhibit the growth of *S. marcescens* isolate at 400 mg/mL (40% concentration). Sub-inhibitory concentrations of xylitol 10% (100 mg/mL) and 5% (50 mg/mL) that are equivalent to 1/4 and 1/8 MICs, respectively, were used for testing the inhibitory activities of xylitol against virulence of *S. marcescens*.

### 3.2. Xylitol Inhibits Biofilm Formation

For biofilm production assessment, the ODc and OD of tested *S. marcescens* isolate were measured, and it was considered strong-biofilm forming as its OD > 4× ODc. Biofilm formation was assayed in the absence and presence of xylitol, and xylitol was found to inhibit biofilm formation. Biofilm inhibition ranged between 74.22% and 81.03% at 5% and 10% of xylitol, respectively (Figure 1B). Microscopic visualization of biofilm under the effect of xylitol was also performed by light microscopy. The microscopic examination revealed significant reduction in both the thickness of and surface area covered by the biofilms in xylitol’s presence (Figure 1A).

### 3.3. Xylitol Decreases Prodigiosin Production

The red intracellular prodigiosin dye is produced by *S. marcescens* under the regulation of QS. Xylitol exhibited a marked inhibitory activity against prodigiosin production that reached 63.42% and 78.47% at concentrations 5% and 10% of xylitol, respectively (Figure 2).

### 3.4. Xylitol Interferes with Protease Activity

Skim milk agar was used to assay the effect of xylitol on protease activity. It was found that xylitol completely abolished protease activity (Figure 3).

### 3.5. Xylitol Inhibits Swimming and Swarming Motilities

Prior to host tissues adherence and formation of biofilms, *S. marcescens* employs swarming and swimming motilities [41]. Xylitol reduced swimming motility by 92.56% at 5% and 94.88% at 10%, and swarming motility by 88.88% at 5% and 93.17 at 10% (Figure 4).

### 3.6. Xylitol Increases the Sensitivity to Oxidative Stress

The hydrogen peroxide disk diffusion assay was used to assess the ability of xylitol to hinder the *S. marcescens* resistance to oxidative stress. In the presence of xylitol, the susceptibility to oxidative stress was enhanced by 16% for xylitol (5%) and 23.72% for xylitol (10%) (Figure 5).

### 3.7. Xylitol Decreased the Expression of S. marcescens Virulence Genes

The inhibitory activities of xylitol against QS and virulence of *S. marcescens* were confirmed genetically by qRT-PCR. When the expression of virulence genes was evaluated in xylitol-treated and untreated *S. marcescens*, it was found that the expression levels of all of *pigP*, *shlA*, *fimA*, *fimC*, *bsmB*, *rssB*, *flhC*, *flhD*, and *rsmA* genes significantly decreased in xylitol-treated culture. The down-regulation of the expression of tested genes was fluctuated between 2- to 4-fold reduction in expression levels as indicated in (Figure 6).

### 3.8. In-Vivo Protection Activity of Xylitol Against S. marcescens

Furtherly, the protective activity of xylitol was in-vivo assessed against *S. marcescens*. Four mouse groups, each comprised of five mice, were assigned and the dead mice were recorded in each group. However, all mice survived in the two negative groups, and only three survived out of five mice in the positive control group. On the other hand, all mice injected with xylitol-treated *S. marcescens* survived (100%); this confers a 40% protection in comparison to the group injected with untreated *S. marcescens*. The survival of mice was recorded over 5 successive days and plotted by the Kaplan–Meier method where significance (*p* < 0.05) was evaluated using the Log-rank test, GraphPad Prism 8 (Figure 7). Clearly, xylitol significantly decreased the *S. marcescens* capacity to kill mice, ascertained by applying the Log rank test for trend (*p* = 0.02).

### 3.9. Xylitol Could Bind to Smar Receptor In-Silico

The binding mode of xylitol was revealed from the performed molecular docking study with SmaR protein. The binding interactions of xylitol and C4-HSL with the target receptor are shown in Figure 8. These interactions included both hydrophobic interaction and hydrogen bonding. The autodock scores for each ligand in addition to the interacting residues are presented (Table 1). Xylitol showed a good ability to bind to SmaR receptors. This might indicate its ability to hinder the binding of the natural ligand to its receptor, and this may result in the inhibition of QS and its regulated virulence factors.

## 4. Discussion

*Serratia marcescens* is an evolving nosocomial pathogen. Due to its ability to adhere to invasive hospital instrumentation and intravenous tubing, its eradication is difficult by conventional methods [42]. That results in causing various healthcare related infections [43]. Among several gram-negative bacteria, *S. marcescens* aquires control over their physiological traits using the QS communication machinery. The QS system in *S. marcescens’* SmaI/SmaR communication system depends on secretion of autoinducers termed homoserine lactones (HSLs). QS orchestrates motility, biofilm formation, production of hydrolytic enzymes, and other virulence factors as lipase, protease, hemolysin, and chitinase [6,44,45,46,47].

Bearing in mind the high antibiotic resistance of *S. marcescens*, it is vital to seek a therapeutic tool other than antibiotics that does not target bacterial growth. Targeting QS can fulfil this purpose because it does not affect cell viability. Instead, QS inhibition can disarm the virulence factors of *S. marcescens*, leaving it eradicable by the immune system [48]. This study was conducted to explore xylitol’s ability to interfere with growth and to curtail the virulence regulated by QS in the emerging nosocomial bacterium *S. marcescens*. Xylitol inhibited *S. marcescens’* growth at 400 mg/mL (40%).

To test the ability of xylitol to interfere with virulence, sub-inhibitory concentrations of xylitol (10% that represents 1/4 MIC and 5% that represents 1/8 MIC) were used for the assays. Xylitol significantly reduced biofilm formation by 74.22% and 81.03% at 5% and 10% of xylitol, respectively. This may be attributed to the anti-adhesive capability of xylitol [49,50,51]. Swarming and swimming were previously linked to microbial cytotoxicity and required for bacterial adherence as the first step of biofilm formation [52,53]. *Serratia marcescens* is motile and can move via swimming and swarming types of motility [54]. Xylitol blocked swimming motility by 92.56% (5% xylitol) and 94.88% (10% xylitol). More or less similar inhibition of swarming motility was achieved (88.88% at 5% and 93.17% at 10%).

The production of the red characteristic pigment prodigiosin was markedly inhibited by xylitol (63.42% at 5% and 78.47% at 10%). This is preliminary evidence of the inhibitory activity of xylitol against QS because it is well reported that prodigiosin is produced under the control of QS [30]. Moreover, xylitol’s ability to interfere with protease activity was investigated by the skim milk agar method, and it could completely inhibit protease activity of *S. marcescens* at both 5% and 10%. Protease activity was reported to have a vital role in human infection and to stimulate inflammatory response [48,55]. Furthermore, resistance to oxidative stress was screened, and xylitol presence enhanced the sensitivity to oxidative stress induced by hydrogen peroxide by 16% (xylitol 5%) and 23.72% (xylitol 10%).

To confirm the activity of xylitol against *S. marcescens* virulence, the expression rate of some genes involved in these traits under the effect of xylitol (5%) was measured by quantitative Realtime PCR. As compared to the control untreated culture, the xylitol-treated culture showed significant down-regulation of all tested genes. *S. marcescens* swarming motility is regulated by flhDC flagellar regulatory operon that encodes the flagellar transcriptional activator FlhD and the flagellar transcriptional regulator FlhC [56,57]. Furthermore, RsmA is considered as a vital component of the swarming regulatory network [41,58,59,60]. Noteworthy, the flhDC regulator is under the control by the RssAB two-component system [59,61]. In the light of the down-regulation of *flhC*, *flhD*, *rssB and rsmA* by xylitol, it can be understood why it blocked swimming and swarming motilities.

Prodigiosin biosynthesis is encoded by the prodigiosin biosynthetic operon pigA-N [62,63]. Xylitol reduced the *pigP* gene expression, which accounts for the inhibition of prodigiosin production. Our study showed that xylitol could downregulate the expression of the *shlA* gene. *S. marcescens* secretes the pore-forming toxin ShlA on host cell-to-cell junctions to help invade the host tissues [64].

To establish infections, bacteria adhere to the tissues employing fimbria or pili whose production is also related to biofilm formation in *S. marcescens* [65]. Type I pili is encoded by the fimABCD operon [66]. Moreover, BsmA and BsmB proteins enhance the production of type I pili in *S. marcescens* [1,66]. The genes *fimA* gene encodes fimbrial A subunit protein, while type 1 fimbriae regulatory protein FimB and BsmB protein are encoded by *fimB* and bsmB genes, respectively. In this study, xylitol downregulated the expression of fimA, *fimC* and *bsmB*. This may represent an explanation for the remarkable biofilm-inhibitory activity of xylitol.

To confirm our results, xylitol’s ability to protect mice against the virulence of *S. marcescens* was performed by the survival test. Xylitol increased the survival of the mice from 60% in the control treated group to 100%. The ability of xylitol to bind to SmaR receptors was analyzed by the docking analysis, and it was found that xylitol could interfere with the binding of the natural ligand C4-HSL to SmaR receptors, resulting in possible QS inhibition, and hence anti-virulence activity.

The use of hypertonic sugars to inhibit growth and virulence was previously reported. In a study on multidrug-resistant *Pseudomonas aeruginosa* clinical isolates, hypertonic glucose inhibited the growth at 30%. Moreover, *P. aeruginosa* motility, biofilm formation, pyocyanin and elastase were significantly blocked by hypertonic glucose (5% and 10%). The expression of *P. aeruginosa* QS genes was down-regulated in glucose hypertonic concentration (20%). Moreover, the administration of hypertonic glucose (20%) increased the survival rates of Galleria mellonella larvae from *P. aeruginosa* [67]. This may suggest a role for the osmotic stress of xylitol in attenuation of the virulence of *S. marcescens*.

## 5. Conclusions

QS regulates the bacterial virulence and its inhibition can attenuate the virulence of bacteria leading to enhancement of the ability of the immune system to remove the bacteria. The advantage of bacterial virulence inhibition that it is less likely to induce resistance. Xylitol may be a good weapon, or at least an adjunctive weapon, to treat infections caused by *S. marcescens* as it showed growth inhibiting activity and virulence attenuating activity, and it may be further considered in management of wound infection by *S. marcescens*. In the current study, we are introducing the application of xylitol and other sugar hypertonic preparations as adjunctive therapy for further investigation on treatment of resistant infections.

## Figures and Tables

**Figure 1 microorganisms-09-01083-f001:**
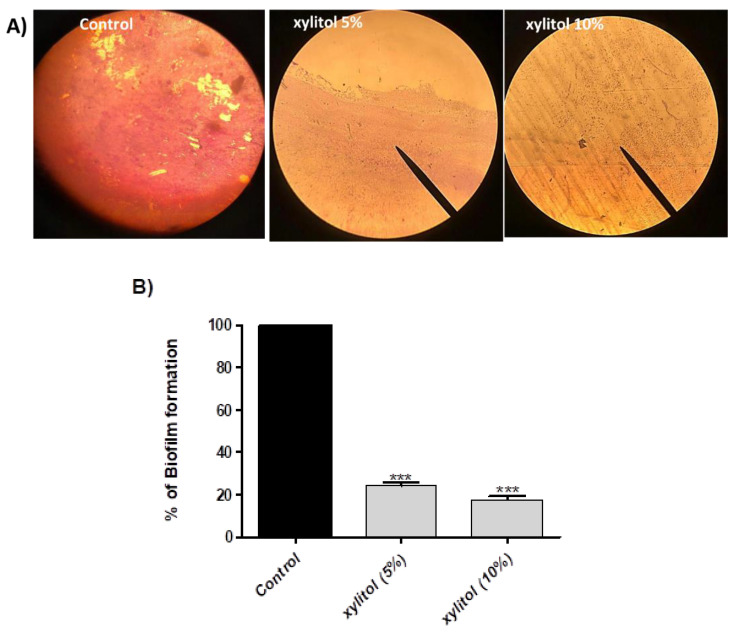
Biofilm inhibition of *S. marcescens* by xylitol. (**A**) Microscopic examination of biofilm inhibition by xylitol. Biofilm was formed on glass cover slips by xylitol-treated and untreated bacteria, and light microscopy was used for visualization. Both thickness and biofilm biomass were reduced by xylitol. (**B**) Biofilm was allowed to form on microtiter plate wells in the presence or absence of xylitol (5% and 10%) and adherent cells were stained with crystal violet, glacial acetic acid was added to dissolve the dye, and the absorbance was measured at 590 nm. The test was performed in triplicate. Biofilm formation was significantly reduced by xylitol. The data shown are the means ± standard errors. One-way ANOVA test followed by Dunnett’s Multiple Comparison test was used for statistical analysis. ***: *p* ≤ 0.001.

**Figure 2 microorganisms-09-01083-f002:**
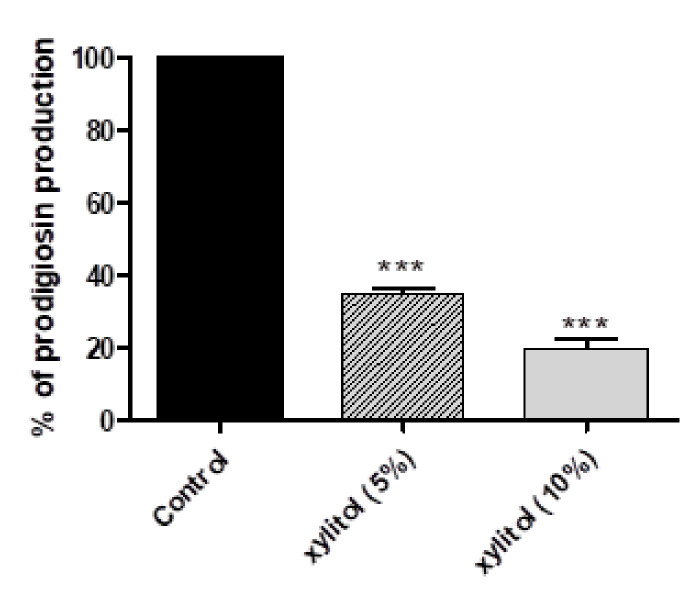
Inhibition of prodigiosin production by *Serratia marcescens* by xylitol. Prodigiosin pigment was extracted from xylitol-treated and untreated cells by acidified ethanol. The test was performed in triplicate and the absorbance was measured at 534 nm. Xylitol significantly reduced prodigiosin production. The data shown are the means ± standard errors. One-way ANOVA test followed by Dunnett’s Multiple Comparison test was used for statistical analysis. ***: *p* ≤ 0.001.

**Figure 3 microorganisms-09-01083-f003:**
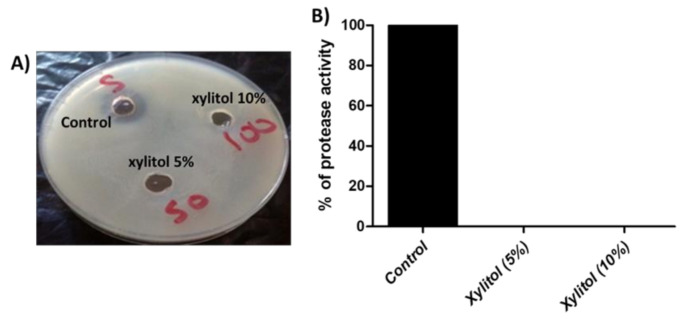
(**A**) Inhibition of protease activity by the skim milk agar method. The wells made in skim milk LB agar plates were inoculated with aliquots of (100 µL) of the supernatants of xylitol-treated and untreated cultures. The clear zones around the wells were measured. The test was performed in triplicate. Xylitol completely blocked the protease activity. (**B**) The data shown are the means ± standard errors. One-way ANOVA test followed by Dunnett’s Multiple Comparison test was used for statistical analysis.

**Figure 4 microorganisms-09-01083-f004:**
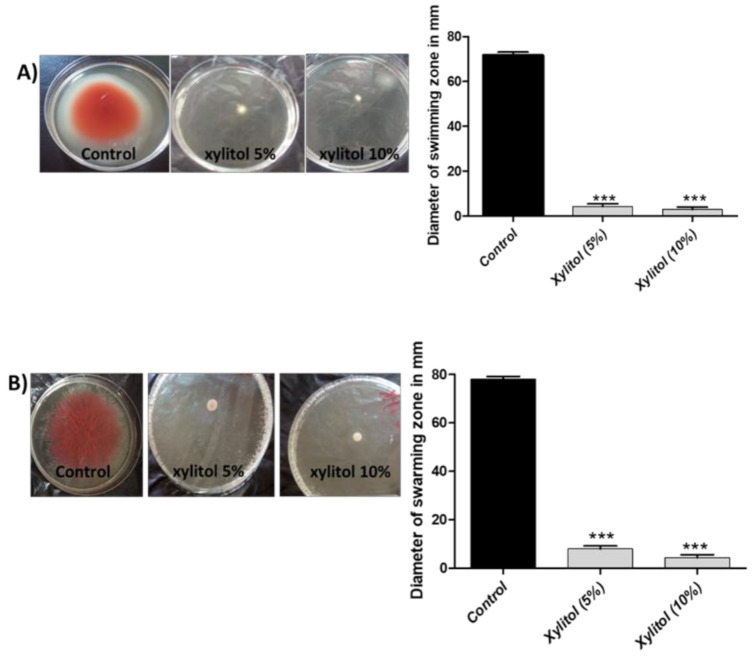
Inhibition of bacterial motility by xylitol. (**A**) Swimming LB agar plates with xylitol and control plates were center stabbed with 5 µL from an overnight culture of *S. marcescens* and the diameter of swimming zones were measured. The test was performed in triplicate. Xylitol significantly reduced swimming motility. The data shown are the means ± standard errors. One-way ANOVA test followed by Dunnett’s Multiple Comparison test was used for statistical analysis. (**B**) Swarming LB agar plates with xylitol and control plates were surface inoculated with 5 µL from an overnight culture of *S. marcescens* and the diameter of swarming zones were measured. The test was performed in triplicate. Xylitol significantly reduced swarming motility. The data shown are the means ± standard errors. One-way ANOVA test followed by Dunnett’s Multiple Comparison test was used for statistical analysis. ***: *p* ≤ 0.001.

**Figure 5 microorganisms-09-01083-f005:**
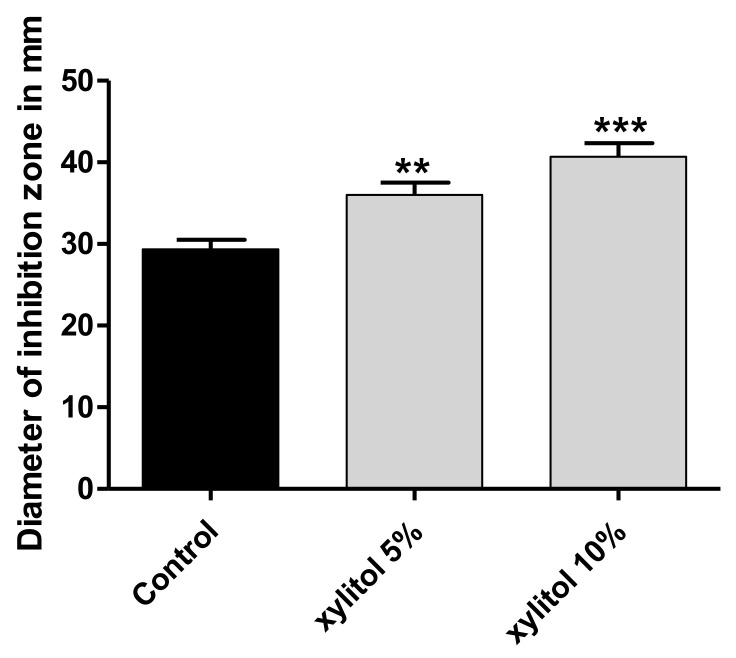
Augmentation of sensitivity to oxidative stress by xylitol. Aliquots of 100 µL of overnight culture of *S. marcescens* were spread on the surface of TSA plates with and without xylitol and 10 µL of 3% hydrogen peroxide were added to a filter-paper disk on the plates. The plates were incubated and the inhibition zones were measured. The test was performed in triplicate. Xylitol significantly increased the inhibition zones and the sensitivity to oxidative stress. The data shown are the means ± standard errors. One-way ANOVA test followed by Dunnett’s Multiple Comparison test was used for statistical analysis. **: *p* ≤ 0.01 and ***: *p* ≤ 0.001.

**Figure 6 microorganisms-09-01083-f006:**
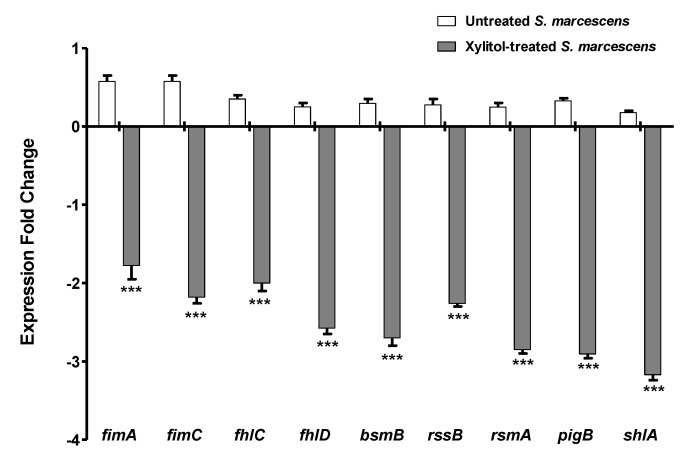
Xylitol decreased the expression of virulence genes of *S. marcescens*. RNA of xylitol-treated and untreated *S. marcescens* cultures was isolated to synthesize cDNA. cDNA was amplified and the change in the expression of each gene were normalized to the rplU gene as the housekeeping gene. The test was performed in triplicate. Xylitol (5%) significantly reduced the expression levels of *fimA*, *fimC*, *flhC*, *flhD*, *bsmB*, *rssB*, *rsmA*, *pigP*, *and shlA* genes. The data shown are the means ± standard errors. One-way ANOVA test followed by Dunnett’s Multiple Comparison test was used for statistical analysis. ***: *p* ≤ 0.001.

**Figure 7 microorganisms-09-01083-f007:**
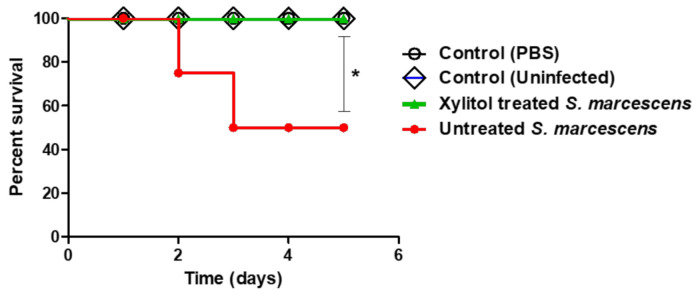
Xylitol protected mice against death by *S. marcescens*. Four mouse groups of healthy mice comprising five mice each were used. In group 1, mice were intraperitoneally injected with xylitol-treated *S. marcescens* in sterile PBS, group 2 was injected with untreated bacteria, group 3 was injected with sterile PBS and group 4 was left un-inoculated. Mouse survival in each group was recorded every day over 5 days, plotted using the Kaplan–Meier method, and significance (*p* < 0.05) was calculated using the Log-rank test, GraphPad Prism 5. All mice in groups 3 and 4 (negative control groups) survived, while only 60% of mice survived (three out of five mice) in the group of the untreated bacteria. In contrast to untreated *S. marcescens*, all mice injected with xylitol-treated *S. marcescens* survived, showing 100% survival, conferring 40% protection (Log rank test for trend * *p* = 0.02).

**Figure 8 microorganisms-09-01083-f008:**
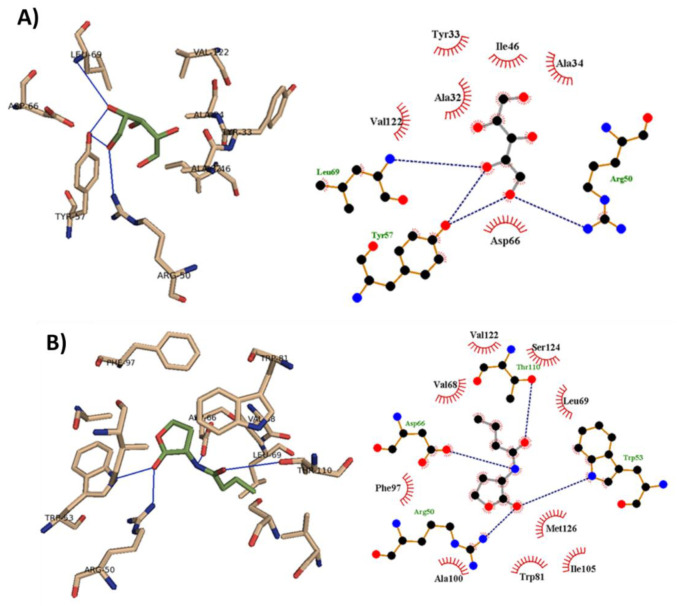
The Molecular docking of (**A**) Xylitol and (**B**) C4-HSL into the active site of SmaR protein 3D representation (left) and 2D Schematic interaction (right). Xylitol could bind with SmaR receptors by hydrogen bonding and hydrophobic interaction.

**Table 1 microorganisms-09-01083-t001:** The binding mode of xylitol and natural ligand with the different residues inside the active site of SmaR protein.

Ligand	H-Bonding	Hydrophobic Interaction	Autodock Score
Xylitol	Arg 50, Tyr 57, Leu 69	Ala 32, Tyr 33, Ala 34, Ile 46, Asp 66, Val 122	−4.5
C4-HSL	Arg 50, Trp 53, Asp 66, Thr 110	Val 68, Leu 69, Trp 81, Phe 97, Ala 100, Ile 105, Val 122, Ser 124, Met 126	−6.4

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
