# Peer review of "Xylitol Inhibits Growth and Blocks Virulence in Serratia marcescens"

_microorganisms, 2021, doi:10.3390/microorganisms9051083_

Round 1

Reviewer 1 Report

The manuscript is interesting as it reports the application of a well-known antimicrobial method on an emerging pathogen. The work reported appear to be novel.

However, there some issues that must be addressed before publication:

  1. Clarify the concentration of xylitol. Is it 5 and 10 % w/w?
  2. The experiments with mice do not make much sense. Xylitol is effective only at high concentrations for wound applications. What is the point of injecting pre-treated cells?
  3. Figure 1: how was the coverage calculated from the microscope figures? And the thickness?
  4. Please explain what is the difference between the three coupons in Figure 3A and how the bars in Figure 3B were calculated. Two bars seem missing
  5. Please show the disc on the agar plates and how the zone of inhibition was measured (Figure 5)

Please italicize the names of microorganisms as appropriate.

Author Response

Dear Reviewer,

We appreciate your valuable and constructive comments and suggestions, which greatly helped us to improve the manuscript. In the light of the provided comments, we revised our manuscript, Please find the attached document "Reply to Reviewer 1"

Bests,

Reviewer 2 Report

In this work, the authors investigate the effect of xylitol in a strain of Serratia marcescens on a number of phenotypes including growth, biofilm formation, quorum sensing, motility and virulence in a mouse model of infection. They show that sub-MIC concentrations of xylitol have an inhibitory effect on those phenotypes. To get an insight into the molecular mechanism of phenotype inhibition, the authors showed in docking studies that xylitol can bind to the SmaR C4-AHL quorum sensing receptor.

General comments

This is a relevant study that shows the effect of a common sweetener on the physiology of a microbe. It would be good to know whether a similar effect of xylitol can be obtained with other strains of S. marcescensfor one or other phenotypes or whether the effect is unique to one strain. As xylitol is a sugar, is it metabolized by S. marcescens. In the opinion of this reviewer, the manuscript needs to be revised to work out the major messages more clearly.

Specific comments

l.26: another?

l.28: Has this been shown for Serratia marcescens?

l.67: In the long run, virulence escape mutants can occur. In Pseudomonas aeruginosa, for example, virulent strains that do not possess the type three secretion system can emerge. My suggestion is to delete this sentence.

l.106: Preferably, the biofilm formation assay should be described in a basic outline.

l.128: …motility…

l.146: to synthesize…

l.156/l.188 ff: at the sub-MIC concentration used, is there still a growth delay? Perhaps it would be informative to show growth curves in the presence and absence of xylitol.

l.189: Are these physiologically relevant concentrations of xylitol? Please put the concentrations used in a context.

l.196: Figure 1A

l.203: For this reviewer it is not clear what is shown on Figure 1A. What is the magnification, a scale bar would help.

l.207 and elsewhere: Triplicate. Biological or technical triplicates? If biological, how many technical triplicates?

l.233: This statement needs a reference.

Author Response

Dear Reviewer,

We appreciate your valuable and constructive comments and suggestions, which greatly helped us to improve the manuscript. In the light of the provided comments, we revised our manuscript, Please find the attached document "Reply to Reviewer 2"

Bests,

Round 2

Reviewer 1 Report

The authors have responded satisfactorily to my comments